# Characterization of New *ATM* Deletion Associated with Hereditary Breast Cancer

**DOI:** 10.3390/genes12020136

**Published:** 2021-01-21

**Authors:** Sandra Parenti, Claudio Rabacchi, Marco Marino, Elena Tenedini, Lucia Artuso, Sara Castellano, Chiara Carretta, Selene Mallia, Laura Cortesi, Angela Toss, Elena Barbieri, Rossella Manfredini, Mario Luppi, Tommaso Trenti, Enrico Tagliafico

**Affiliations:** 1Center for Genome Research, University of Modena and Reggio Emilia, 41125 Modena, Italy; sandra.parenti@unimore.it (S.P.); claudio.rabacchi@unimore.it (C.R.); 2Centre for Regenerative Medicine, University of Modena and Reggio Emilia, 41125 Modena, Italy; chiara.carretta@unimore.it (C.C.); selene.mallia@unimore.it (S.M.); rossella.manfredini@unimore.it (R.M.); 3Department of Medical and Surgical Sciences, University of Modena and Reggio Emilia, 41124 Modena, Italy; sara.castellano@unimore.it (S.C.); mario.luppi@unimore.it (M.L.); 4Department of Laboratory Medicine and Pathology, Diagnostic Hematology and Clinical Genomics Unit, Modena University Hospital, 41124 Modena, Italy; marino.marco@aou.mo.it (M.M.); tenedini.elena@aou.mo.it (E.T.); artuso.lucia@aou.mo.it (L.A.); t.trenti@ausl.mo.it (T.T.); 5Department of Oncology and Hematology, University Hospital of Modena, 41124 Modena, Italy; cortesi.laura@aou.mo.it (L.C.); angela.toss@unimore.it (A.T.); barbieri.elena@aou.mo.it (E.B.)

**Keywords:** next-generation sequencing, breast cancer, *ATM*, homologous recombination repair, hereditary cancer syndromes, clinical genomics, molecular diagnostics

## Abstract

Next-generation sequencing (NGS)-based cancer risk screening with multigene panels has become the most successful method for programming cancer prevention strategies. *ATM* germ-line heterozygosity has been described to increase tumor susceptibility. In particular, families carrying heterozygous germ-line variants of *ATM* gene have a 5- to 9-fold risk of developing breast cancer. Recent studies identified *ATM* as the second most mutated gene after *CHEK2* in BRCA-negative patients. Nowadays, more than 170 missense variants and several truncating mutations have been identified in *ATM* gene. Here, we present the molecular characterization of a new *ATM* deletion, identified thanks to the CNV algorithm implemented in the NGS analysis pipeline. An automated workflow implementing the SOPHiA Genetics’ Hereditary Cancer Solution (HCS) protocol was used to generate NGS libraries that were sequenced on Illumina MiSeq Platform. NGS data analysis allowed us to identify a new inactivating deletion of exons 19–27 of *ATM* gene. The deletion was characterized both at the DNA and RNA level.

## 1. Introduction

Breast cancer is one of the most common tumors in the female population, and 5–10% of cases arise as hereditary breast cancer (HBC) [1]. The assessment of the hereditary cancer risk that can be obtained following the mutational study with next-generation sequencing (NGS)-based multigene panels can allow the choice of the best risk prevention program and has become the most successful method for programming cancer prevention [2,3,4]. Although *BRCA1* and *BRCA2* remain the most frequently mutated genes, mutations in other DNA repair genes account for a significant fraction (~25%) of HBC cases [5]. *ATM* germ-line heterozygosity occurs in about 1% of the population and has been described to increase tumor susceptibility. In particular, families carrying heterozygous germ-line variants of *ATM* gene have a 5- to 9-fold risk of developing breast cancer, particularly in women younger than 50 years [6]. Approximately, 3% of the families affected by hereditary breast and ovarian cancer harbors a loss-of-function *ATM* mutation [7]. Recent studies identified *ATM* as the second most mutated gene after *CHEK2* in BRCA-negative patients [8]. In particular, the cumulative risk of breast cancer by age 50 was estimated for *ATM* heterozygous mutation carriers to be 6.02%, while by age 80, was estimated to be 32.83% [9]. Nowadays, more than 170 missense variants and several truncating mutations have been identified in *ATM* gene [8]. Few data are available on the deletions and rearrangements of the *ATM* gene, especially since not all analytical pipelines of the NGS data implement specific algorithms for predicting CNVs. The artificial intelligence-based NGS data analysis platform, SOPHiA DDM software, incorporates a specific algorithm to study CNVs from NGS data generated using the SOPHiA Genetics’ Hereditary Cancer Solution (HCS) kit. Here, we present the molecular validation and characterization both at DNA and RNA level of a new *ATM* deletion identified with diagnostic NGS for hereditary cancer risk screening.

## 2. Materials and Methods

### 2.1. Subjects

Written informed consent was obtained from all subjects. Total genomic DNA was extracted from blood samples using Maxwell 16 Blood DNA Purification Kit and Maxwell 16 MDx AS3000 instrument (Promega, Madison, WI, USA).

### 2.2. NGS Sequencing

Two hundred nanograms of genomic DNA has been processed with the Hereditary Cancer Solution (HCS) kit (SOPHiA GENETICS, Saint-Sulpice, Switzerland). The capture-based target enrichment of 26 cancer related genes (*ATM*, *APC*, *BARD1*, *BRCA1*, *BRCA2*, *BRIP1*, *CDH1*, *CHEK2*, *EPCAM*, *FAM175A*, *MLH1*, *MRE11A*, *MSH2*, *MSH6*, *MUTYH*, *NBN*, *PALB2*, *PIK3CA*, *PMS2*, *PMS2C*, *PTEN*, *RAD50*, *RAD51C*, *RAD51D*, *STK11*, *TP53*, *XRCC2*) and the library construction protocols were carried out exclusively with the automated procedure implemented on the STARlet platform (Hamilton Company, Reno, NV, USA). Library quantification was carried out with fluorometric quantitation using Qubit dsDNA High Sensitivity kit (Thermofisher Scientific, Waltham, MA, USA). Quality control was performed by analyzing the profile of each sample obtained with Bioanalyzer DNA 1000 (Agilent Technologies, Santa Clara, CA, USA).

Sequencing was achieved onto a 600-cycle format V3 flow-cell, via Illumina MiSeq DX (Illumina, San Diego, CA, USA) platform according to their and SOPHiA Genetics’ protocols. Sequencing data were processed for single nucleotide variants (SNVs), indels, and copy number variations (CNVs) via the SOPHiA DDM platform based on SOPHiA Artificial Intelligence (AI).

### 2.3. Multiplex Ligation-Dependent Probe Amplification (MLPA) Analysis

One hundred nanograms of genomic DNA of the proband was analyzed for CNVs detection in the *ATM* gene with SALSA MLPA P041 and P042 probemix, (MRC Holland, Amsterdam, The Netherlands) according to manufacturer’s instructions on ABIPrism 3130XL Genetic Analyzer (Thermo Fisher Scientific). The data were analyzed with Coffalyzer software (MRC Holland).

### 2.4. Long-Range PCR of the Gene Region Including the Deletion

The deletion, detected by NGS sequencing and MLPA, was confirmed with long-range PCR using the Fast Start PCR Master Mix (Roche, Penzberg, Germany). FastPCR Software (Primer Digital Ltd., Helsinki, Finland) was used to design the following primers: 

*ATM*-e18-4FW: 5′-GTGTAACTACTGCTCAGACCAA-3′

*ATM*-e28-8R: 5′-CAGACCAATACTGTGTCCTTTAGGGCA-3′

DNA was amplified using the following PCR conditions: an initial denaturation at 94 °C for 3′, followed by 10 cycles at 94 °C for 20″, 59 °C for 30″, 68 °C for 7′, and by other 20 cycles at 94 °C for 20″, 59 °C for 30″ and 68 °C for 7′ with an increase of 10″ for every PCR cycle, and a final extension at 68 °C for 13′.

### 2.5. Characterization of the Deletion Breakpoint

The PCR product was sequenced with ABIPrism 3130XL Genetic Analyzer (Thermo Fisher Scientific), using the following primers:

*ATM*-e18-10FW: 5′-CTACCAAATCCCTCCACCTGCAT-3′

*ATM*-E28-7R: 5′-GAGCAGGATCCAAATCCCTAACAGAGT-3′

The nucleotide sequence was compared to wild-type *ATM* genomic sequence [LRG_135, NG_009830.1]. Intronic repeated sequences were found with Repeat Masker software (http://www.repeatmasker.org/cgi-bin/WEBRepeatmasker).

### 2.6. Analysis of ATM mRNA

Total RNA was extracted with Maxwell 16 Total RNA Purification Kit and Maxwell 16 MDx AS3000 instrument (Promega). In total, 500 ng of total RNA was reverse transcribed using MMLV Reverse Transcriptase (Thermo Fisher Scientific) as previously described [10]. PCR amplification of the gene region including the deletion was performed as described above. Then, the PCR fragment harboring the deletion was extracted from agarose gel and sequenced as previously described [11]. The nucleotide sequence of the amplified cDNA fragment was compared to the wild-type sequence of *ATM* mRNA [NM_000051.3].

## 3. Results

### 3.1. Identification of Genomic Variant by NGS Analysis

The proband, belonging to a family of Italian ancestry, had been referred for genetic testing based on a family history of breast cancer at early onset. She received a diagnosis of unilateral breast cancer HER2+, ER+ and PR− at age 34. Her mother was affected by unilateral hormone receptors positive (HER2+, ER+, PR+) breast cancer, diagnosed at the age of 45 years, with metastases in axillary lymph nodes. The proband was previously tested for the mutational status of only *BRCA1* and *BRCA2* genes and reported as negative for pathogenic variants. In order to identify any further variants that could explain the phenotype, the proband was analyzed using the CE-IVD Hereditary Cancer Solution (HCS) assay by SOPHiA Genetics. The results revealed a wide heterozygous deletion of exons from 19 to 27 of *ATM* gene.

### 3.2. MLPA Analysis and Breakpoint Characterization

In order to confirm the deletion of the region spanning from exon 19 to exon 27 of the *ATM* gene, MLPA analysis was performed on the genomic DNA of the proband with two different mixes of probes in order to examine the whole gene. The results confirmed this deletion in heterozygous state (Appendix A). To further characterize the deletion, we used a long-range PCR design to amplify the breakpoint region in proband DNA and, both in her mother and sister, for verifying the segregation. This assay confirmed the presence of the deletion, showing a 2.5 Kb PCR product in proband and in her mother (Figure 1). In order to explain the possible causal mechanism of the deletion, we analyzed the sequence using the Repeat Masker software for the identification of interspersed repeats and low complexity DNA sequences. The output revealed several repetitive genomic elements, as ALU sequences, in particular in intron 18. To characterize in detail the exact breakpoint, we sequenced the PCR amplicon by Sanger sequencing, revealing a small six-nucleotide element, GGCTCA, overlapping the two introns involved in the rearrangement (Figure 2) and allowing the accurate description of the deletion at DNA level (c.2838+2162_4110-292del).

### 3.3. Characterization of the mRNA Sequence

To characterize the RNA transcript derived from the deleted allele, we reverse-transcribed the proband’s mother and sister RNA. Unfortunately, the proband’s RNA was no longer available. The cDNA was then amplified and the gene region carrying the deletion was sequenced. As shown in Figure 3A, the mother cDNA originated both the wild-type allele (1500 bp) and the mutated one (200 bp). In the mutated isoform, exon 28 is joined to exon 18 (Figure 3B). This variant disrupts the reading frame of the mRNA generating a premature stop codon predicted to encode for a truncated protein of 952 aa, instead of the wild-type 3056 aa ATM protein (p.Tyr947Glyfs*7).

## 4. Discussion

The characterization of the mutational status of genes involved in homologous recombination repair pathway is absolutely the most important approach for designing the prevention strategies for inherited cancer syndromes. A large number of missense and truncating *ATM* variants have been detected in *BRCA1*- or *BRCA2*-negative patients [8].

SOPHiA Genetics HCS gene panel allowed us to detect a novel *ATM* CNV in a BRCA-negative HBC patient. The deletion of exons 19 to 27 of the *ATM* gene was detected in an Italian woman who received a diagnosis of unilateral breast cancer at the age of 34. Her mother had unilateral receptor-positive breast cancer, at the age of 45, with axillary lymph node involvement. The NGS detected CNV was firstly validated by MLPA. Afterward, long-range PCR and Sanger sequencing were used in order to characterize the breakpoint at DNA level (c.2838+2162_4110-292del) in proband and to study the segregation also in her mother and in her sister.

A further characterization at RNA level on the proband’s mother and sister identified the presence of both the wild-type and the mutant allele in the mother’s sample. In the mutated isoform, exon 28 is joined to exon 18, disrupting the mRNA reading frame and generating a premature stop codon. The predicted aberrant ATM protein consists of 952 amino acids instead of 3056 amino acids of wild-type ATM. This abnormal protein lacks the domain involving the amino acid positions 1373–1382, required for c-Abl protein interaction and crucial to mediate cell cycle arrest in G1 phase [12]. In addition, at least three other important domains are deleted from the ATM protein, such as FAT (FRAP-*ATM*-TRRAP), PIKK (phosphatidylinositol 3-kinase-related kinase domain) and FATC (FAT C-terminal domain) domains, mediating the majority of ATM functions [13,14].

## Figures and Tables

**Figure 1 genes-12-00136-f001:**
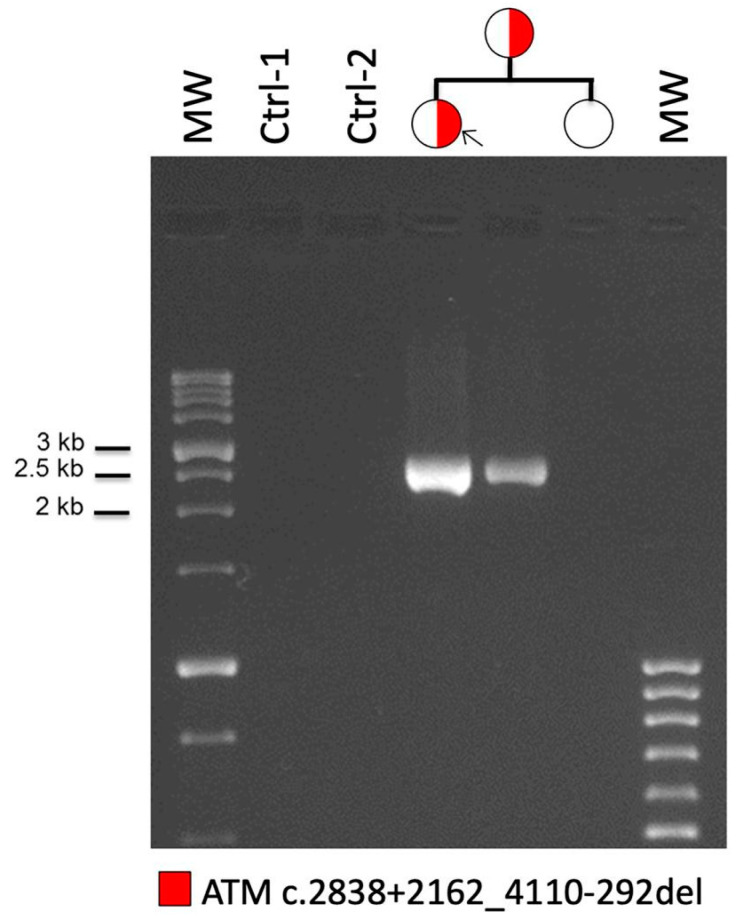
Breakpoint characterization of *ATM* exon 19-27 deletion: long-range PCR confirmed the presence of the deletion, showing a 2.5 Kb PCR product in proband DNA. The analysis of other family members confirms the presence of the deletion in proband’s mother (lanes 4 and 5). Two DNAs from healthy subjects were amplified as control (lanes 2 and 3). Molecular weight (MW) markers are loaded in lanes 1 and 7.

**Figure 2 genes-12-00136-f002:**
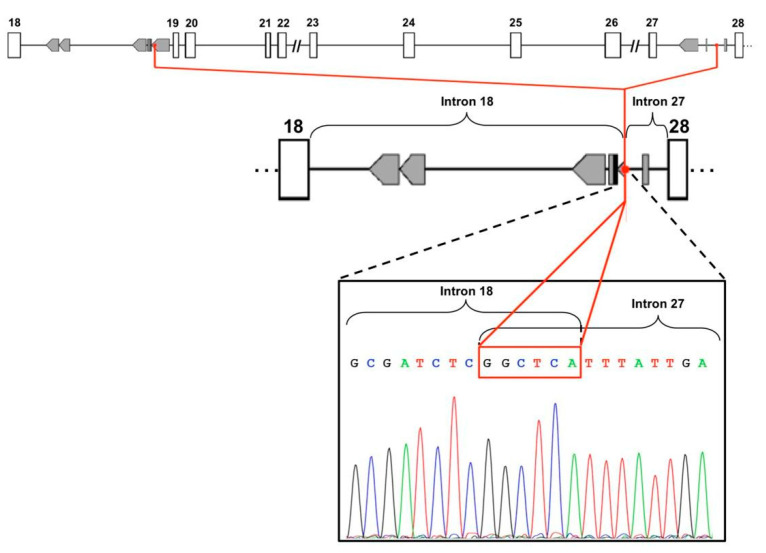
Analysis of repetitive genomic elements and Sanger sequencing: upper and middle part of the figure show the analysis of genomic sequences performed through Repeat Masker software that revealed several repetitive genomic elements, in particular ALU sequences (in gray boxes), in intron 18. Lower part of the figure, Sanger sequencing of the 2.5 Kb PCR product shows the breakpoint region of the rearrangement, characterized by a small six-nucleotide element, GGCTCA, overlapping the two introns.

**Figure 3 genes-12-00136-f003:**
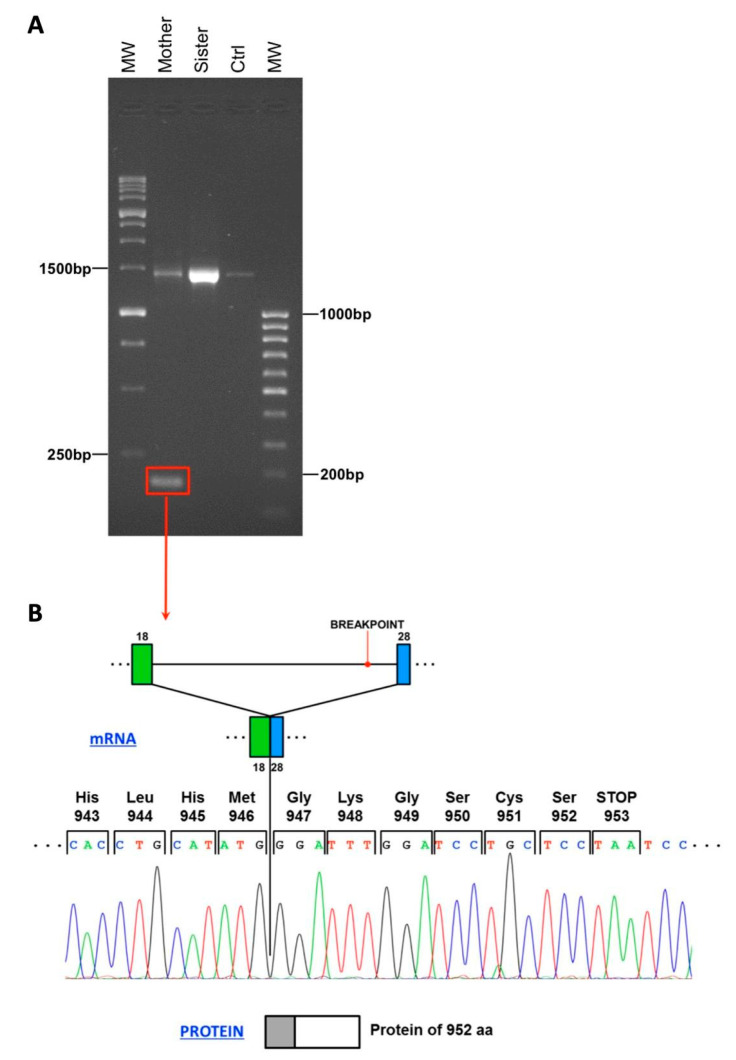
(**A**) Characterization of mRNA sequence resulting from the deletion: RNA derived from proband’s mother showed a 200 bp PCR product represented in Panel A that underwent Sanger sequencing, displayed in Panel B. Proband’s sister shows only the wild-type form of the transcript. (**B**) This analysis allowed us to demonstrate that this variant generates a premature stop codon that encodes for a truncated protein of 952 aa (grey box), instead of the wild-type 3056 aa ATM protein (grey+white boxes).

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
