# Peer review of "Characterization of New ATM Deletion Associated with Hereditary Breast Cancer"

_genes, 2021, doi:10.3390/genes12020136_

Round 1

Reviewer 1 Report

The manuscript is of sufficient interest, the results are reported clearly and the discussion is thorough. I recommend a general review of English. However, the writing style is clear and understandable.

Author Response

The manuscript is of sufficient interest, the results are reported clearly and the discussion is thorough. I recommend a general review of English. However, the writing style is clear and understandable.

Thank you so much for the revision. We revised English language as best we could

Reviewer 2 Report

Introduction, paragraph 1, line 37: Could the authors describe what they mean by programming cancer prevention strategies? I find this sentence difficult to interpret as written. 

Introduction, paragraph 1, line 42: Could the authors provide cumulative risk estimates for developing breast cancer by, for example, 80 years of age, in the setting of an ATM PV.

Introduction, paragraph 1, line 44-45: Could the authors provide references to support the statement: "Recent studies identified ATM as the second 44 most mutated gene after CHEK2 in BRCA-negative patients."

Results, paragraph 3.1, line 108: Is the family associated with this pedigree of Italian ancestry? Please clarify.

Results, paragraph 3.1, line 110: This should be re-phrased to "hormone receptor positive." Was the proband's mother ER+HER2-? Do the authors have information on HER2 status of both the proband and mother?

Results, paragraph 3.1: I would recommend that the authors include information around the proband being diagnosed with unilateral, ER+ breast cancer at 34 years old in the results; the first mention of this is in the figures and discussion. 

Results & Discussion, throughout: When the authors write monolateral breast cancer, do they mean unilateral breast cancer?

Discussion, line 176: Although suspected, have the authors performed any functional or structural analysis to confirm if this variant and the aberrant ATM protein is highly likely to be pathogenic? This would be an important area of further study.

Author Response

Introduction, paragraph 1, line 37: Could the authors describe what they mean by programming cancer prevention strategies? I find this sentence difficult to interpret as written. 

Thank you for your suggestion. We have rephrased the sentence to make it clearer: “the assessment of the hereditary cancer risk that can be obtained following the mutational study with Next Generation Sequencing (NGS) based multigene panels can allow the choice of the best risk prevention program and has become the most successful method for programming cancer prevention”.

Introduction, paragraph 1, line 42: Could the authors provide cumulative risk estimates for developing breast cancer by, for example, 80 years of age, in the setting of an ATM PV.

Following the referee's suggestion, we added the cumulative risk of breast cancer by age 50, and 80 and added a specific reference.

Introduction, paragraph 1, line 44-45: Could the authors provide references to support the statement: "Recent studies identified ATM as the second most mutated gene after CHEK2 in BRCA-negative patients."

Thanks again for the note, the reference is the ref.# 8.

Results, paragraph 3.1, line 108: Is the family associated with this pedigree of Italian ancestry? Please clarify.

Right observation: in the revised version of the manuscript, we specified that the proband belongs to a family of Italian ancestry

Results, paragraph 3.1, line 110: This should be re-phrased to "hormone receptor positive." Was the proband's mother ER+HER2-? Do the authors have information on HER2 status of both the proband and mother?

Following the referee's suggestion, we re-phrased the sentences specifying exactly the positivity for HER2 and for the ER and PR receptors for proband and for her mother

Results, paragraph 3.1: I would recommend that the authors include information around the proband being diagnosed with unilateral, ER+ breast cancer at 34 years old in the results; the first mention of this is in the figures and discussion. 

Thank you: it has been done

Results & Discussion, throughout: When the authors write monolateral breast cancer, do they mean unilateral breast cancer?

Yes of course, it is a mistake. We corrected the mispelling

Discussion, line 176: Although suspected, have the authors performed any functional or structural analysis to confirm if this variant and the aberrant ATM protein is highly likely to be pathogenic? This would be an important area of further study.

Here we characterize the ATM mutation at genomic DNA and RNA level only and did not perform any functional or structural studies for this case report. Further studies could be useful to better understand the pathogenic effect. The ATM variant has been classified as pathogenic according to recommendations of the American College of Medical Genetics and Genomics and the Association for Molecular Pathology (ACMG/AMP) consensus [Richards S. et al. Genet Med 2015, 17:405-423]

Reviewer 3 Report

Parenti et al. report their characterization of a novel ATM deletion which they claimed to be associated with hereditary breast cancer. They present a new ATM deletion, identified with the CNV algorithm implemented using NGS analysis. It is an inactivating deletion of exons 19-27 of the ATM gene. The presence of this deletion was confirmed at the DNA and RNA level in a single subject. 

Although their identification of this novel ATM deletion is presented well and has scientific merit, its importance is diminished by the lack of a molecular studies that can provide evidence to their claims. Specifically, how deletion of exon 19-27 on a single ATM allele affects transcriptional and cellular programs to result in carcinogenesis in breast tissue. In addition, the lack of other subjects with this particular mutation and similar clinical outcomes is a major drawback to the significance of their results.

Specific questions that need to be addressed:

Is the mutant truncated ATM protein described in line 152 produced or expressed in a cellular context at all? Does it have a cellular function? Can this function be linked to carcinogenesis in breast tissue?

Following the discussion on lines 177-178, how is the recruitment of c-Abl blocked? Would there be a compensatory mechanism that allows for a higher expression of the normal ATM allele? 

Author Response

Is the mutant truncated ATM protein described in line 152 produced or expressed in a cellular context at all? Does it have a cellular function? Can this function be linked to carcinogenesis in breast tissue?

We characterized the ATM mutation at genomic DNA and RNA level only and did not study the protein expression or synthesis. The predicted mutant protein is about 31% of the normal one, lacking several functional domains. This is just a case report describing the molecular characterization of a new ATM deletion at both the DNA and RNA levels. We have not done any studies on the protein, therefore, even if it is very likely that ATM protein is profoundly altered by the effect of the deletion, we have no data to confirm if this mutant protein has a specific function or if it is synthetized and immediately degraded due to, for example, incorrect folding. Further studies could be useful to better understand the pathogenic effect.

Following the discussion on lines 177-178, how is the recruitment of c-Abl blocked? Would there be a compensatory mechanism that allows for a higher expression of the normal ATM allele? 

The paper cited in the reference section (ref. #12) about the interaction of ATM with c-abl demonstrates that SH3 domain of tyrosine kinase c-Abl interacts with a proline-rich DPAPNPPHFP motif (residues 1,373- 1,382) of ATM. The mutation described in this paper creates a frameshift starting from aa 946 and impairs the generation of the c-Abl binding motif. This interaction is important during DNA damage response. Although Scott et.al (Scott SP, Bendix R, Chen P, Clark R, Dork T, Lavin MF. Missense mutations but not allelic variants alter the function of ATM by dominant interference in patients with breast cancer. Proc Natl Acad Sci U S A. 2002 Jan 22;99(2):925-30.) demonstrated that missense variants of ATM are expressed at the same level as the wild-type one, our described deletion generates a very small protein. For this reason, we cannot speculate that the same conclusion is applicable to our case. Further studies will be necessary in order to evaluate the stability of this protein, but, since this work is meant to be a case report, this analysis is beyond the scope of the paper.

Round 2

Reviewer 3 Report

None of the previous issues raised are addressed by their revisions. The study is still lacking and based on their comments they are of their study's shortcomings. More experiments are needed to support their conclusion.